# Mixed-methods exploration of the knowledge of young adults about blood donation processes; a one-center cross-sectional study in a tertiary institution

Belinda Baidoo*[☯], Elizabeth Ankomah[☯], Mohammed Alhassan, Godfred Benya, Emmanuella Obike[ID], Audrey Benfo, Joseph Boachie, Patrick Adu[ID]*

Department of Medical Laboratory Sciences, School of Allied Health, Sciences College of Health and Allied Sciences, University of Cape Coast, Cape Coast, Ghana

☯ These authors contributed equally to this work.

* Patrick.adu@ucc.edu.gh (PA); belindabaidoo1206@gmail.com (BB)

**Data Availability Statement:** All relevant data are within the paper and its Supporting Information files.

## Abstract

Ghana is a majority youthful population, but is only able to meet 60% of its annual blood donation requirements. Although tertiary students in Ghana may serve as important blood donor resource by virtue of their higher educational attainment, data about their blood donation processes-specific knowledge are scarce. This study therefore explored the perspectives, and experiences of young adults regarding blood donation processes. This exploratory study employed mixed-methods approach (semi-structured questionnaire and focus group discussion, [FGD]). Data collection was sequential; the questionnaire distribution was completed before FGD commenced; themes that emerged from the questionnaire responses guided FGDs. Convenience sampling technique was used to recruit 382 young adults (15–49 years). All statistical analyses were undertaken using the two-tail assumptions; $p<0.05$ was considered statistically significant. Majority (79.3%) of the participants were in their twenties, with only 1.3% being 40–49 years old. Although two-thirds of participants expressed willingness to donate blood, less than a-third (31.7%; 127/382) had previously donated blood. Overall, less than one-third of participants could correctly identify the minimum weight (26.4%), or the inter-donation interval (14.7%); 37.4% and 58.1% could respectively indicate the required donor age or ≥3 infectious agents screened for prior to blood collection. Among previous donors, 37.2%, 28.1% and 43.0% could identify the required weight, acceptable inter-donation period, and donor age respectively. Two-thirds and a-third of participants preferred voluntary unrelated, and paid donations respectively. Whereas 42.4% of participants indicated intrinsic health benefits of blood donation, 17.0% suggested that blood donation was associated with disease risks. Both previous donors and non-donor groups considered lack of education, fear of post-donation health issues and lack of privacy at blood collection centers as main hindrances to donor recruitment. Targeted intentional blood donation-specific educational campaigns are warranted to address the blood donation processes knowledge gap among the study population.

**Funding:** The author(s) received no specific funding for this work.

**Competing interests:** The authors have declared that no competing interests exist.

**Abbreviations:** FGD, Focus group discussio; SSA, sub-Saharan Africa; WHO, World Health Organization; NBSG, National Blood Services of Ghana.

## Introduction

Blood transfusion is an integral part of the healthcare delivery system in sub-Saharan Africa (SSA). The high transfusion burden is occasioned by the high anaemia prevalence due to road traffic accidents, surgical procedures, high prevalence of falciparum malaria infections, and haemoglobinopathies in SSA. In accordance with recommendations from the World Health Organization (WHO), 10 blood donations per 1000 population (1% donation rate) is sufficient to supply the needs of the respective population [1,2]. Whereas the industrialized countries are able to achieve 50 donations per 1,000 population (5%) which is in excess of the recommended 1% per population, developing countries are only able to achieve 3 donations per 1,000 population (0.3%), which is inadequate for blood transfusion needs. This disproportional blood donation rate is reflected in the concentration of 48% of the global blood transfusions in the industrialized countries with 15 percent of the world's population [2]. Across Africa, it is estimated that more than 40 countries fail to meet the WHO's donation goal of 10 units per 1000 people [1,3]. Consequently, an estimated one-fourth of peripartum maternal deaths in SSA have been attributed to inaccessible blood transfusion [4]. Additionally, delayed transfusion has also been linked to increased mortality incidence in paediatric cases diagnosed with malaria-associated anaemia [5]. Paradoxically, the developing countries however have a majority youthful population compared to the ageing population in the developed countries. All things being equal, this overly youthful population should potentially translate into adequacy of donor blood stocks. Presently, the Ghana healthcare system is only able to approximately meet 60% of its annual blood transfusion needs [6]. Previous studies in the SSA region have found altruism [7,8], a sense of communal collectivistic outlook [9,10] and reciprocity of blood donation [11,12] as factors that could be harnessed to motivate the populace to increase blood donation rate per a given population.

However, a previous cross-sectional study among the general populace in the Northern region of Ghana found that approximately one-third of blood donors knew the minimum age for blood donation or blood donation frequency per year [13] suggesting poor knowledge in blood donation-specific processes. Other studies elsewhere have also similarly indicated that education might be an important consideration in the drive towards achieving donor blood sufficiency [14–17]. Tertiary students within the SSA context, by virtue of national representation as well as meeting donor age eligibility remains a potential resource in efforts to maximize blood donations across the country [18]. When sufficiently motivated, these tertiary students could be permanent voluntary, non-remunerated blood donors as well as potentially serve as champions/ambassadors of blood donor mobilization drives within their respective communities. Hypothesizing that the endemic apathy towards blood donations could be a function of educational attainment, we sought to explore the perspectives of young adults in a tertiary educational setting to assess their knowledge regarding the blood donation requirements and procedures. Our overarching aim therefore, was to identify the perceived challenges that these youth face regarding voluntary blood donation as well as identify actionable points that could be harnessed to improve blood donations rate per 1000 population to improve access to blood products across the country.

## Material and methods

### Characteristics of the qualitative enquiry

This study adopted the postpositivist interpretive framework in which the enquiry was approached as a series of logically related steps to achieve multiple perspectives to the data collection. Specifically, a semi-structured questionnaire was initially used to collect quantitative

data from participants; subsequently, a focus group discussion was used to collect qualitative data from participants who had consented during the initial semi-structured questionnaire filling.

## Study site and population

The study was done at the University of Cape Coast (UCC), Cape Coast in the Central region of Ghana. The duration of the study was from August 17, 2022 to December 30, 2022. At the time of the study, the University of Cape Coast had the highest number of tertiary students in the Cape Coast metropolis. The school had a total population of 74,710 of which 18,949 were regular undergraduate students, 1,445 were sandwich undergraduate students, 1,014 regular postgraduate students, 2,773 sandwich postgraduate students, 48,989 distance undergraduate students and 1,540 post graduate distance students.

## Inclusion/Exclusion criteria

Regular undergraduate and post graduate UCC students were included in this study.

Sandwich postgraduate or undergraduate students, or students on the distance learning stream, workers (junior and senior staff), and non-student population in the school environs were excluded from this study.

## Sampling

**Sampling method.** The recruitment of participants was done based on convenient sampling technique.

**Sample size.** The sample size that was used in this study was determined using the Andrew Fisher's Formula.

$$\text{Sample size} = \frac{z - Score^2 \; standard \; deviation(1 - Standard \; Deviation)}{confidence \; interval^2}$$

Parameters used were: confidence interval = 5%, confidence level = 95%, standard deviation = 0.5, Z-score based on 5% confidence interval = 1.96.

Sample size $= \frac{(1.95)^2 x 0.5(1-0.5)}{0.05^2} = 384.16$ which was approximated to 385.

However, to cater for potential non-return of questionnaire, a total of 424 questionnaires (10% allowance) were distributed to students. Overall, 382 questionnaires were returned (response rate of 90.9%).

**Sampling of participants.** Participants were recruited from their hostels, hall of residence, lecture rooms and library. This selection was done to get students from all fields of study to prevent skewed data collection. Also, this selection was done taking into consideration regular post graduate and regular undergraduate students who were the target for this study.

**Research instrument.** Two instruments were used for the data collection; semi-structured questionnaires (a quantitative data collection technique) and focus group discussions (a qualitative data collection technique).

**Data collection.** The schematic of the data collection is presented in **Fig 1**.

## Semi-structured questionnaire

Semi-structured questionnaires were used to gather baseline data on participants' knowledge, attitudes, misconceptions and challenges about blood donation. The semi-structured questionnaire had four parts. Part 1 solicited socio-demographic background data, as well as an option to opt-in for the focus group discussion. Part 2 solicited information on participants'

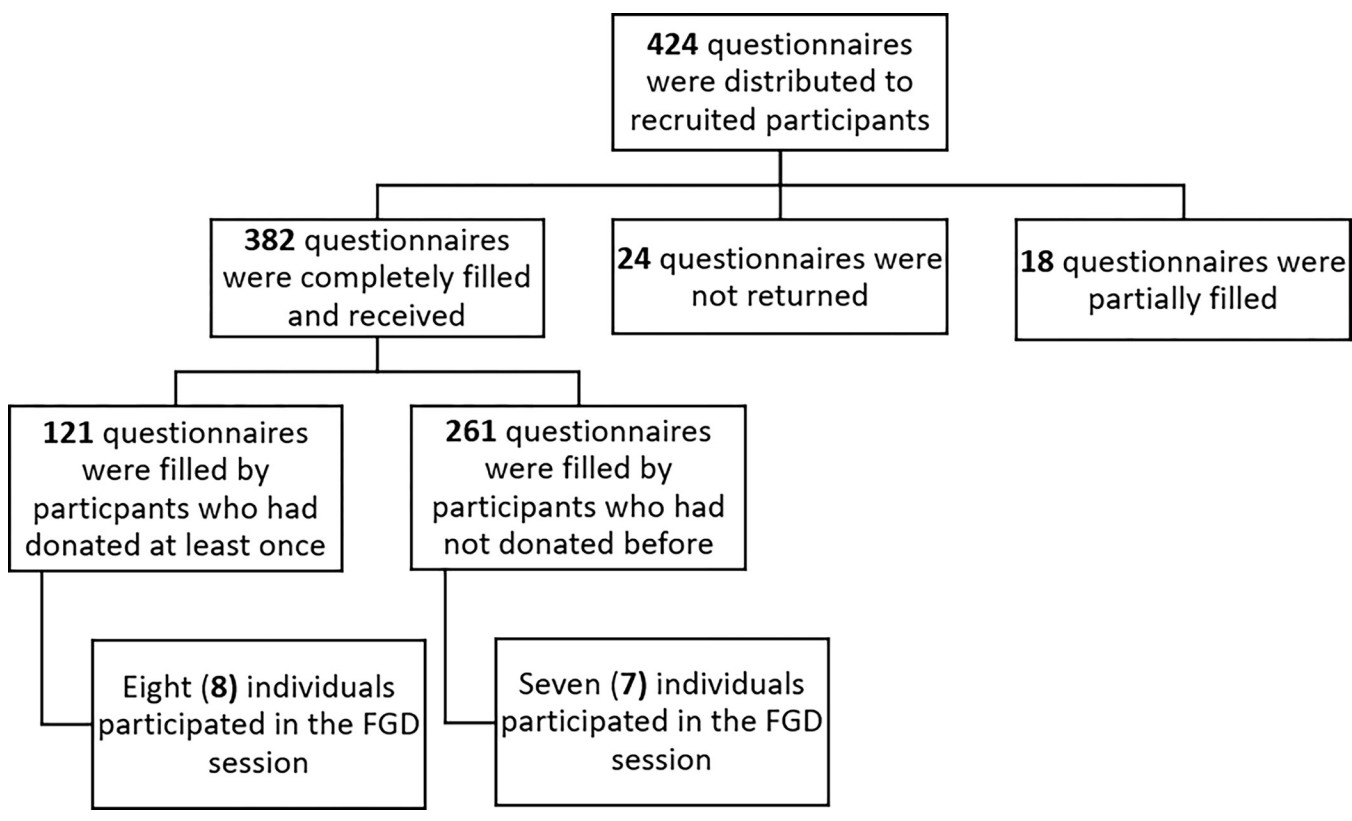

**Fig 1. A schematic of the procedures for the data collection.** [FGD: Focus group discussion].

knowledge on blood donation. Part 3 examined participants' attitude towards blood donation and part 4 explored participants' perspectives on the challenges to blood donation. In the part 3 and 4 of the questionnaire that examined attitudes and challenges towards blood donation respectively, participants judged their agreement with specific statements on a five-point Likert scale ranging from 1 (strongly disagree) to 5 (strongly agree). The raw questionnaire data has been enclosed as a supporting information (S2 File).

## Focus group discussion

The FGD was the second phase of the data collection (the FGD guide is enclosed as S1 File). The inclusion criterion for the FGD was that a participant must have completed the initial questionnaire data collection phase and must have voluntarily consented to becoming part of the FGD; invitations were extended to only such individuals for the FGD. Since the themes that informed the FGD guide emerged from the preliminary analyses of the questionnaire responses, only individuals who had answered the questionnaires could meaningfully participate in the FGD. Each FGD session lasted about 60 minutes and was conducted in English language since all the participants understand and speak English language fluently; however, participants were encouraged to switch between local dialect and English in instances where they believed doing so would enable them to express themselves unhindered. All FGDs were facilitated by trained research assistants and the supervisors who also had local dialect competencies.

In total, 15 individuals took part in the FGD and each session included both females and males. Two separate FGDs were undertaken: one for participants with prior blood donation

experience (8 individuals), and another session for participants who had never donated blood in their lifetime (7 individuals). Participants in the focus group discussion were encouraged to give their opinion on the questions that were posed to the group based on their personal experiences and community interactions. If they did not wish to answer or were uncomfortable with any of the questions or take part in any aspect of the discussions, they were encouraged to say so. All FGDs took place at the School of Allied Health Science conference room I (SAHS--Room I), University of Cape Coast. The FGD guide was adapted from a previous study [19] to obtain responses to research questions. The key areas of the FGD guide were (a) participants' understanding of blood donor eligibility criteria, (b) what participants considered to be motivators for blood donations (c) lived experiences of individuals who had previously donated blood (d) perceived barriers/challenges to blood donations. All these areas were further explored based on participants responses; the transcription of the FGDs has been enclosed as a supporting information (S1 File).

All five (5) research assistants (three females & two males) were trained in the ethics guiding the moderation and facilitation of FGDs. Prior to the start of data collection, the FGD guide was piloted among some medical laboratory students, university of Cape Coast. The rationale for the piloting was to provide experiential training in FGDs to the research assistants, assess responses to the questions, and modify the FGD guide based on responses obtained. All FGDs were audio recorded with permission from the participants. For the purpose of the discussion, participants were addressed by their first names.

## Data analysis

Quantitative data was analyzed and presented as proportions using Statistical Package for Social Science (SPSS) version 20 as a statistical tool. In the data analyses presented in Table 3, the strongly agree and agree categories were merged as agree, whereas strongly disagree and disagree were also merged as "disagree" to ultimately provide three scales i.e., disagree, uncertain, and agree. The themes that emerged from the questionnaire data analysis served as the guide for the two (2) focus group discussions. Data obtained from the focus group discussion was thematically analyzed to identify common themes that have been reported in this study.

The FGDs were transcribed verbatim by trained personnel with prior experience in transcribing qualitative interviews. Transcriptions were done in accordance with previously suggested best practices in qualitative research interviews [20]. The quality and accuracy of the interview transcription were independently assessed by two members of the research team who listened to the recorded interviews vis-a-viz the interview transcription. Thematic analyses of the transcripts were independently undertaken by two members of the research team, after an initial generation of a priori list for organizing themes based on the research objectives. The coding and analyses were in accordance with published guidelines for thematic data analysis [21]. In the result section, the major themes and the corresponding sub-themes under each major theme are presented in figures.

## Statement of ethics and human subjects issues

Ethical approval was obtained from the University of Cape Coast Institutional Review Board (ethical clearance ID: UCCIRB/CHAS/2022/77). Our study protocol conformed to the provisions of the Declaration of Helsinki in 1995 (as revised in Brazil 2013) including confidentiality, risks and benefits assessments, consent to participate, and ensuring respect to participants. It was ensured that written informed consent was sought from participants and that the information given was kept confidential. Informed consent was acquired by explaining the research to the participants to understand before engaging them; since the age of consent is 18 years in

Ghana, introductory letters were sent to obtain accent from the parents of participants who were less than 18 years old. It was also ensured that participants were not coerced into participating but joined voluntarily and that the evaluation did not cause any harm to the participants. The anonymity of participants was ensured by not recording information in a way that links participants' responses with identifying information. Moreover, participants had the right to withdraw their consent and discontinue participation from the study at any time. The information obtained from the research was used for the intended purpose only. Although participants who consented to participating in the FGD were initially required to provide phone numbers, these were removed from the data after the FGDs to maintain data anonymity.

## Results

The socio-demographics of the participants are presented in Table 1. The majority of the participants were males (61.5%) or in their twenties (79.3%). A slight majority of the participants were from the college of humanities, whereas overwhelming majority were single (86.9%) or undergraduate students (87.4%). Overall, less than one-third of the participants had ever donated blood. Of the participants with prior blood donation experience, 24.8% (30/121),

**Table 1. The socio-demographics of study participants.**

| Variable | Frequency, N (%) |
|---|---|
| **Age (years)** | |
| 15–19 | 42 (11.0) |
| 20–29 | 303 (79.3) |
| 30–39 | 32 (8.4) |
| 40–49 | 5 (1.3) |
| **Sex** | |
| Female | 147 (38.5) |
| Male | 235 (61.5) |
| **Marital status** | |
| Cohabiting | 4 (1.0) |
| Divorced | 20 (5.2) |
| Married | 24 (6.3) |
| Single | 332 (86.9) |
| Widowed | 2 (0.5) |
| **College** | |
| CANS | 76 (19.9) |
| Education | 60 (15.7) |
| CoHAS | 119 (31.2) |
| Humanities | 127 (33.3) |
| **Study level** | |
| Undergraduates | 334 (87.4) |
| Masters | 43 (11.3) |
| Doctoral | 5 (1.3) |
| **Have you donated blood before?** | |
| No | 261 (68.3) |
| Yes | 121 (31.7) |

CoHAS: College of Health and Allied Sciences; CANS: College of Agriculture and Natural Sciences.

19.8% (24/121), and 53.7% (65/121) considered themselves as first time donors, family replacement donors and permanent donors respectively.

The study participants were explored per their knowledge about the blood donation screening procedures (Table 2). Whereas a little over half (58.1%) of the participants could correctly identify at least three of the transfusion transmitted infections (TTIs) that are screened for during the pre-donation screening, only 36.0% (80/222) of these had previously donated blood. Also, 5.8% of participants identified malaria screening as a routine test during the pre-donation testing algorithm. Of the 26.4% of the participants who could correctly state the minimum weight required to donate blood, 44.6% (45/101) had previously donated blood. Also, of the 37.4% of the participants who could correctly identify age range for blood donation, only one-third (36.4%; 52/143) had previously donated blood. Whereas a higher proportion of males (39.1% vs 19.7% females) had previously donated blood, a slight majority of participants who had had previous blood donation experience were from the college of health and allied sciences (41.2%). Furthermore, only 28.1% (34/121) of participants with previous blood donation experience could correctly identify the inter-blood period as being 3–5 months.

Participants' attitude towards blood donation were explored per gender and college that a participant belonged to (Table 3). Overall, only one-third of the participants indicated that donating blood saves lives; significantly higher proportion of females believed blood donation

**Table 2. Exploration of data per participants previous blood donation experience.**

| Variable | Total | Have you donated blood before? | | p-value |
|---|---|---|---|---|
| | | No | Yes | |
| **Correctly identify disease screened for prior to donation** | | | | **0.016** |
| ≥3 diseases | 222 (58.1) | 142 (54.4) | 80 (66.1) | |
| 1–2 diseases | 75 (19.6) | 53 (20.3) | 22 (18.2) | |
| None | 63 (16.5) | 53 (20.3) | 10 (8.3) | |
| Malaria | 22 (5.8) | 13 (5.0) | 9 (7.4) | |
| **Correctly identify minimum weight required to donate blood** | | | | **0.001** |
| No | 281 (73.6) | 205 (78.5) | 76 (62.8) | |
| Yes | 101 (26.4) | 56 (21.5) | 45 (37.2) | |
| **What is the acceptable inter-donation period for whole blood?** | | | | **<0.001** |
| ≤2 months | 25 (6.5) | 12 (4.6) | 13 (10.7) | |
| 3–5 months | 56 (14.7) | 22 (8.4) | 34 (28.1) | |
| 6–12 months | 40 (10.5) | 11 (4.2) | 29 (24.0) | |
| >12 months | 5 (1/3) | 3 (1.1) | 2 (1.7) | |
| No idea | 256 (67.0) | 213 (81.6) | 43 (35.5) | |
| **Gender** | | | | **<0.001** |
| Female | 147 (38.5) | 118 (80.3) | 29 (19.7) | |
| Male | 235 (61.5) | 143 (60.9) | 92 (39.1) | |
| **Correctly identify age range required to donate blood** | | | | 0.128 |
| No | 239 (62.6) | 170 (65.1) | 69 (57.0) | |
| Yes | 143 (37.4) | 91 (34.9) | 52 (43.0) | |
| **College** | | | | 0.059 |
| CANS | 76 (19.9) | 54 (71.1) | 22 (28.9) | |
| Education | 60 (15.7) | 43 (71.7) | 17 (28.3) | |
| CoHAS | 119 (31.2) | 70 (58.8) | 49 (41.2) | |
| Humanities | 127 (33.2) | 94 (74.0) | 33 (26.0) | |

The data is presented as frequencies n (%) with statistical significance at p<0.05 determined using chi-square test.

**Table 3. Participants' attitude about blood donation.**

| # | Total | Sex | | p-value | College | | | | p-value |
|---|---|---|---|---|---|---|---|---|---|
| | | Female | Male | | CANS | Education | CoHAS | Humanities | |
| **Blood donation save lives** | | | | **0.001** | | | | | 0.916 |
| Agree | 127 (33.2) | 60 (40.8) | 67 (28.5) | | 25 (32.9) | 22 (36.7) | 36 (30.3) | 44 (34.6) | |
| Uncertain | 14 (3.7) | 10 (6.8) | 4 (1.7) | | 3 (3.9) | 1 (1.7) | 4 (3.4) | 6 (4.7) | |
| Disagree | 241 (63.1) | 77 (52.4) | 164 (69.8) | | 48 (63.2) | 37 (61.7) | 79 (66.4) | 77 (60.6) | |
| **I am comfortable donating blood to unrelated individual** | | | | 0.383 | | | | | 0.619 |
| Agree | 242 (63.7) | 89 (60.5) | 153 (65.7) | | 47 (62.7) | 37 (61.7) | 82 (68.9) | 76 (60.3) | |
| Uncertain | 89 (23.4) | 40 (27.2) | 49 (21.0) | | 16 (21.3) | 17 (28.3) | 22 (18.5) | 34 (27.0) | |
| Disagree | 49 (12.9) | 18 (12.2) | 31 (13.3) | | 12 (16.0) | 6 (10.0) | 15 (12.6) | 16 (12.7) | |
| **Blood donors must be paid when they donate blood** | | | | 0.165 | | | | | 0.695 |
| Agree | 115 (30.1) | 36 (24.5) | 79 (33.6) | | 25 (32.9) | 15 (25.0) | 39 (32.8) | 36 (28.3) | |
| Uncertain | 138 (36.1) | 58 (39.5) | 80 (34.0) | | 26 (34.2) | 19 (31.7) | 44 (37.0) | 49 (38.6) | |
| Disagree | 129 (33.8) | 53 (36.1) | 76 (32.3) | | 25 (32.9) | 26 (43.3) | 36 (30.3) | 42 (33.1) | |
| **Donating blood to a family member is the best** | | | | 0.426 | | | | | 0.406 |
| Agree | 214 (56.2) | 79 (53.7) | 135 (57.7) | | 39 (51.3) | 35 (58.3) | 61 (51.7) | 79 (62.2) | |
| Uncertain | 95 (24.9) | 42 (28.6) | 53 (22.6) | | 20 (26.3) | 11 (18.3) | 35 (29.7) | 29 (22.8) | |
| Disagree | 72 (18.9) | 26 (17.7) | 46 (19.7) | | 17 (22.4) | 14 (22.6) | 22 (18.6) | 19 (15.0) | |
| **Donating blood gives the donor health benefits** | | | | 0.790 | | | | | **<0.001** |
| Agree | 162 (42.4) | 61 (41.5) | 101 (43.0) | | 28 (36.8) | 18 (30.0) | 70 (58.8) | 46 (36.2) | |
| Uncertain | 131 (34.3) | 49 (33.3) | 82 (34.9) | | 34 (44.7) | 21 (35.0) | 29 (24.4) | 47 (37.0) | |
| Disagree | 89 (23.3) | 37 (25.2) | 52 (22.1) | | 14 (18.4) | 21 (35.0) | 20 (16.8) | 34 (26.8) | |
| **Every healthy person should donate blood** | | | | 0.115 | | | | | 0.399 |
| Agree | 201 (52.6) | 71 (48.3) | 130 (55.3) | | 40 (52.6) | 31 (51.7) | 72 (60.50) | 58 (45.7) | |
| Uncertain | 102 (26.7) | 48 (32.7) | 54 (23.0) | | 19 (25.0) | 15 (25.0) | 27 (22.7) | 41 (32.3) | |
| Disagree | 79 (20.7) | 28 (19.0) | 51 (21.7) | | 17 (22.4) | 14 (23.3) | 20 (16.8) | 28 (22,0) | |
| **Blood donation is associated with increased risk of acquiring disease by the donor** | | | | 0.200 | | | | | 0.184 |
| Agree | 65 (17.0) | 31 (21.1) | 17 (22.4) | | 17 (22.4) | 14 (23.3) | 13 (10.9) | 21 (16.5) | |
| Uncertain | 113 (29.6) | 44 (29.9) | 21 (29.4) | | 21 (27.6) | 18 (30.0) | 32 (26.9) | 42 (33.1) | |
| Disagree | 204 (53.4) | 72 (49.0) | 132 (56.2) | | 38 (50.0) | 28 (46.7) | 74 (62.2) | 64 (50.4) | |

The data is presented as frequencies n (%) with statistical significance at p<0.05 determined using chi-square test. CANS: College of Agricultural and Natural Sciences; CoHAS: College of Health and Allied Sciences.

saved lives compared to males (40.8% vs 28.5%; p = 0.001). In relation to who the potential recipient of the donated blood might be, 63.7% of the participants indicated willingness to donate blood to unrelated individuals, although a little over half of participants (56.2%) indicated that blood donation to a relative represents the best option. Moreover, when asked about the possibility of blood donors being paid for donations, participants were equally split between agreeing, disagreeing and being indetermined about such potential payment. Furthermore, less than half of the participants (42.4%) indicated that blood donation was associated with health benefits to the donor compared to 17% who stated that blood donation increased risk of acquiring diseases by the blood donor. A slight majority of participants indicated that every healthy individual should consider donating blood. When participants' attitude towards blood donation were however explored per colleges of students, the responses were comparable except when asked about the health benefits of blood donation where participants from health and allied sciences college comprised a significant proportion.

**Table 4. Exploring the perceptions of participants with previous blood donation experience.**

| Variable | N (%) |
|---|---|
| **There was enough privacy at the blood donation center to discuss my misunderstandings** | |
| Agree | 58 (48.3) |
| Uncertain | 13 (10.8) |
| Disagree | 49 (40.8) |
| **All my questions were adequately answered by the blood donation staff** | |
| Agree | 76 (63.3) |
| Uncertain | 27 (22.5) |
| Disagree | 17 (14.2) |
| **All the blood donation procedures were explained to me** | |
| Agree | 73 (60.8) |
| Uncertain | 17 (14.2) |
| Disagree | 30 (25.0) |
| **I will actively encourage other people to donate blood** | |
| Agree | 91 (75.8) |
| Uncertain | 20 (16.7) |
| Disagree | 9 (7.5) |
| **I will donate blood again once given the opportunity** | |
| Agree | 80 (66.7) |
| Uncertain | 26 (21.7) |
| Disagree | 14 (11.7) |
| **The inconvenience of my previous blood donation process will prevent me from donating blood again** | |
| Agree | 22 (18.3) |
| Uncertain | 29 (24.2) |
| Disagree | 69 (57.0) |
| **The blood donation center staff were very friendly** | |
| Agree | 102 (85.0) |
| Uncertain | 11 (9.2) |
| Disagree | 7 (5.8) |

Table 4 explored the perceptions of a sub-section of the participants who had previously donated blood. Whereas less than half (48.3%) of the participants were satisfied with privacy at the blood collection center, majority (63.3%) were satisfied with the adequacy of answers to their respective questions by the blood collection staff. Also, whereas 18.3% of the previous donors suggested unwillingness to donate blood because of their prior blood donation experience, about two-thirds (66.7%) were willing to donate blood again. Moreover, whereas six-out-of-ten were satisfied with explanation of the blood donation procedures, 75.8% indicated willingness to serve as ambassadors to encourage others to donate blood. Overwhelmingly, the staff at the blood collection centers were considered very friendly in their approach towards the blood donors.

## Thematic analysis of perceived barriers by participants with no previous blood donation

The participants who had never previously donated blood were asked to indicate what they considered to be barriers in the blood donation process. The self-disclosed barriers were thematically analysed and presented in Fig 2. Codes were developed from the responses of the

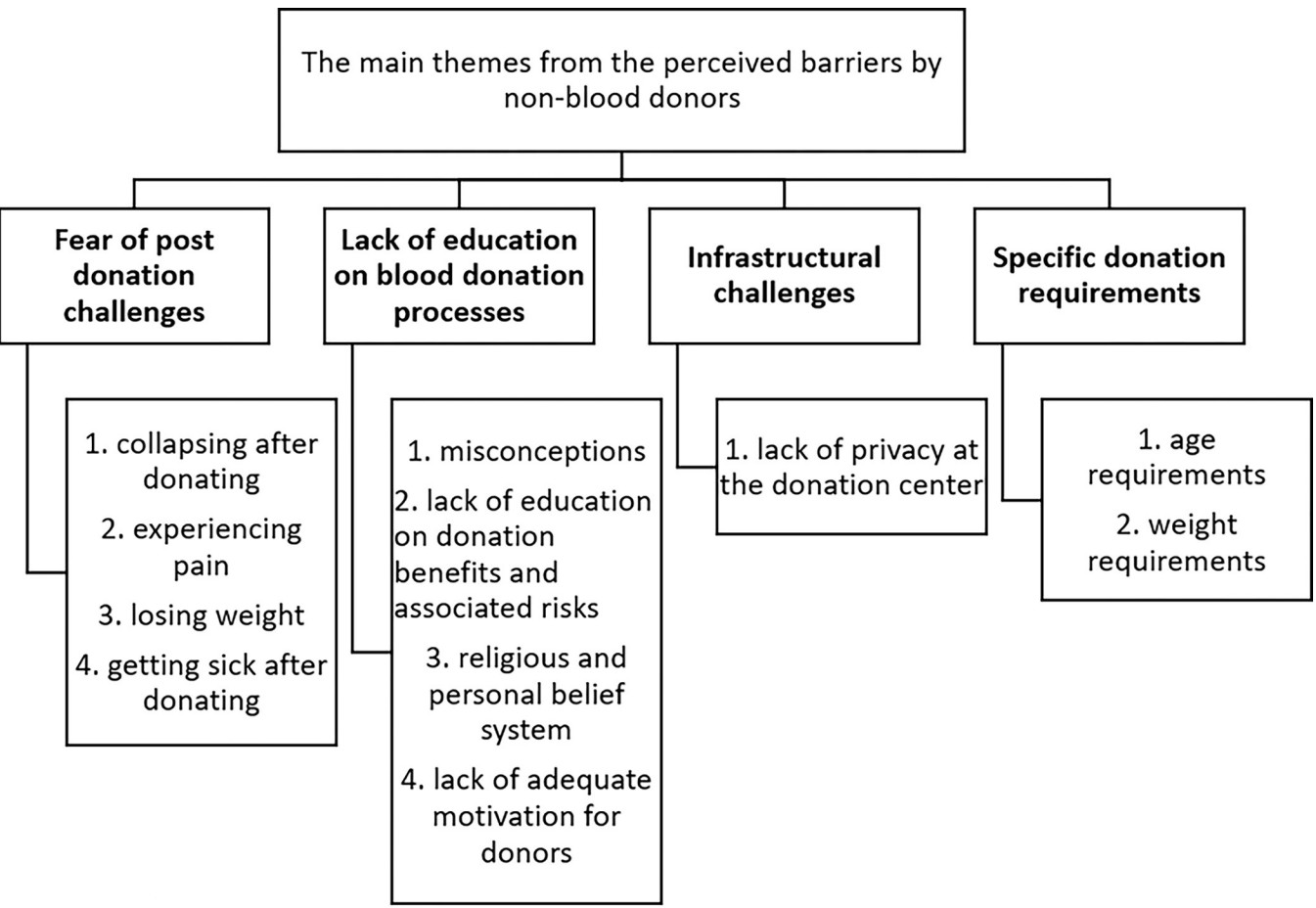

**Fig 2. Thematic analysis of the self-reported barriers experienced by non-blood donors.** The major themes are in bold face type; sub-themes under each major theme are numbered.

participants and grouped into four main themes; fear of post-donation challenges, lack of education on blood donation processes, perceived infrastructural challenges and specificity of blood donation requirements.

### Thematic analysis of barriers experienced by participants with prior blood donation experience

The participants, who had previously donated blood, were asked to indicate what they considered to be barriers in the blood donation process. The self-disclosed barriers were thematically analysed and presented in Fig 3. Codes were developed from the responses of the participants and grouped into three main themes; lack of education on blood donation processes, post-donation issues, perceived infrastructural challenges.

### Discussion

Until laboratory production of blood cells will be fully optimized and scaled up for production to meet global needs, the traditional blood donation process will continue to be an integral part of the healthcare system. The World Health Organization (WHO) recommends that blood donations at 1% of any given population is enough to supply the transfusion needs of

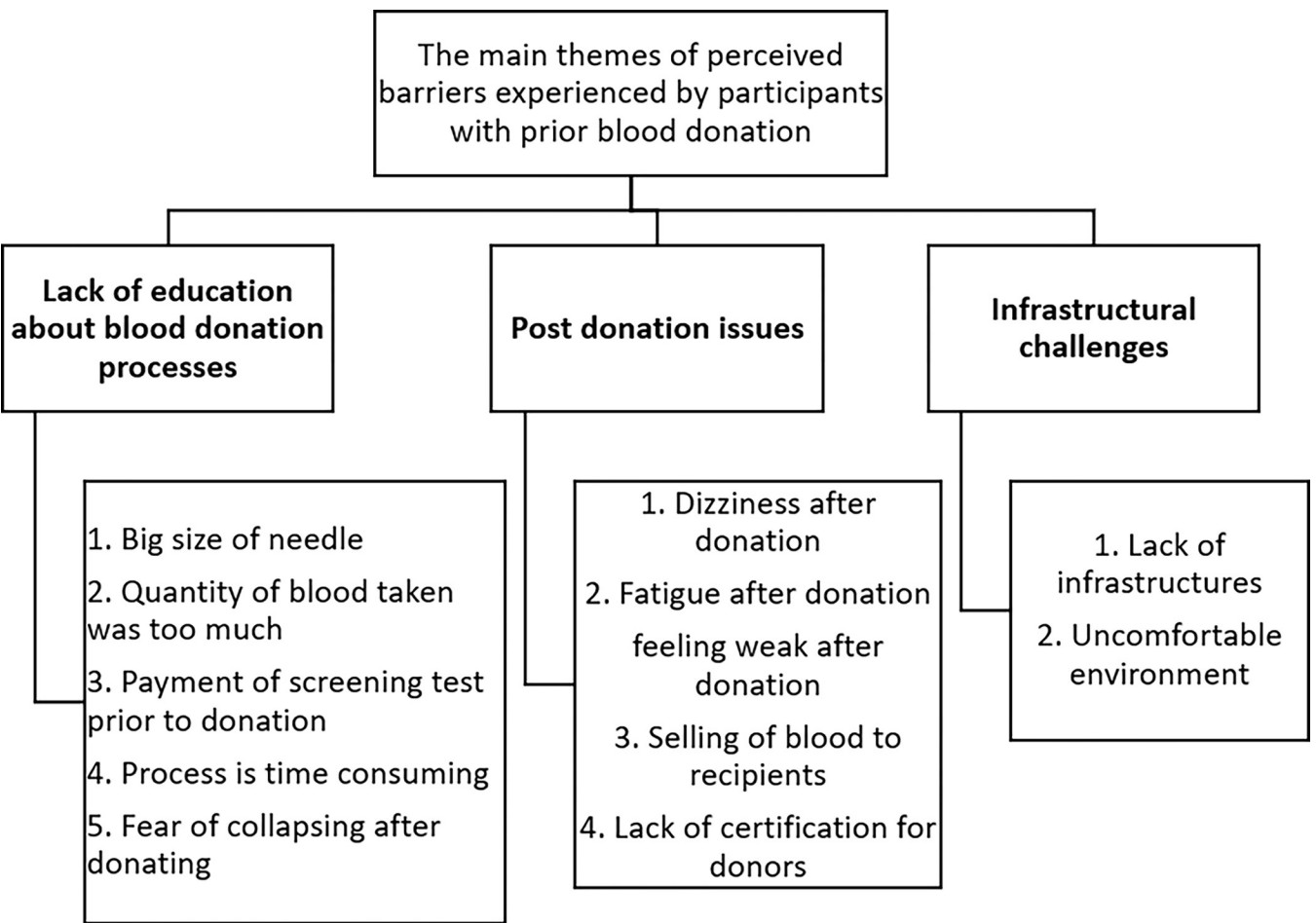

**Fig 3. Thematic analyses of self-reported barriers to blood donation by participants with prior blood donation experience.** The three major themes are in bold face type; the sub-themes are numbered.

that population [2]. In spite of a majority youthful population, sub-Saharan Africa is only able to meet 41.5% of its annual transfusion needs and is consequently plagued with perennial blood shortages [22]. This calls for an urgent need to explore the potential hindrances/challenges associated with blood donation processes in order to institute measures to improve blood donation rate per respective population in sub-Saharan Africa. In Ghana, the National Blood Services published that the country could only meet 60.4% of its blood needs in 2018 [6], in spite of the majority of the population being within the eligible blood donor age (17–60 years). Hypothesizing that educational attainment might be a function of an individuals' willingness to commit to blood donation, we sought to explore the perceptions of young adults within a tertiary institution regarding blood donation practices. Surprisingly, less than one-third (31.7%) of the participants had previously donated blood [23], and the general knowledge about the blood donation eligibility criteria as well as blood donations processes were considerably poor irrespective of a participants' prior blood donation status. Our findings highlight an urgent need for a national public health educational campaign since even those with tertiary education do not necessarily have improved knowledge about the processes involved in blood donor selection as well as the unique role of blood donation within the healthcare delivery.

It is worth stating that the National Blood Services of Ghana have been adopting strategies to sensitize the Ghanaian populace to blood donation. However, our findings are suggestive that either the medium employed is not optimal or not appropriate for the population or the messages are not permeating to the target group in the population. For example, although it is concerning that generally, only one-third of the study population could correctly identify specific items on the blood donation eligibility criteria such as minimum weight required (26.4%), acceptable inter-donation interval (14.7%), eligible age of blood donation (37.4%), what was even more alarming was the fact that even among individuals who have previously donated blood, approximately 40% could correctly identify these blood donor eligibility parameters. Taken together with previous studies [13,24], this brings into question the adequacy of the education that prospective blood donors receive at various blood collection centers prior to blood donation process. It is expected that adequacy of the pre-donation education would cater for a holistic understanding of blood donation processes among those with prior blood donation experience which will equip them to become more inclined to serve as champions of blood donation in their respective communities. However, if these previous donors are oblivious of the criteria that was used to evaluate their eligibility to donate blood, then the adequacy of the donor screening protocols ought to be examined. It is noteworthy that since the University is an equal opportunity institution offering admission to students from all the 16 regions across Ghana, the experiences of these individuals with prior blood donation may have a nationally representative outlook, and may not necessarily represent a single center experience. When participants were asked about the TTIs screened for prior to blood donation, six-in-ten of those with previous blood donation experience compared to five-in-ten of those with no prior blood donation experience could correctly identify ≥3 of the TTI agents. Evidently, this calls for a need to re-evaluate the existing blood donor recruitment protocols within the blood services and perhaps re-strategize with various stakeholder groups to ensure that the educational materials and processes are situated in the Ghanaian cultural context. Any such future endeavour should take cognizance of the diverse ethnic dialects to attain the needed saturation to impart the required behavioural changes. Decidedly, there is a disconnect between what these previous donors ought to have known, and what they actually knew about blood donation protocols since six-out-of-ten of previous donors thought that all the blood donation procedures were explained to them as well as all the question they had about the blood donation processes were answered satisfactorily by the blood collection center staff, even though they could not correctly answer blood donation-related questions. What is encouraging however, was that overwhelming majority of our previous blood donors indicated that the staff at the blood collection center were friendly which may be indicative that empowering these staff with the needed logistics and refresher training in sound blood donor recruitment and retention procedures may improve blood stocks across the country [13,25]. Nearly 7 out of ten of our previous donors indicated willingness to donate blood if given the opportunity. Although intention to donate may not necessarily translate into 100% actual donation [26], it is indicative that some of these previous donors are latent and only need re-activation through targeted campaigns/messages to get them to re-donate blood.

Misconceptions about the blood donation processes were real issues among both participants with no previous blood donation experience and those without prior blood donation experience. In the focus group discussions, whereas participants with no prior blood donation experience expressed grave concerns about perceived weight loss [27,28], getting sick [28], and collapsing post blood donation [29,30], individuals with prior blood donation experience were however concerned about selling of donated blood by staff of blood collection center [31], quantity of blood drawn per donation, big needle size and easy fatigability post blood donation. These misconceptions have been previously reported by other studies from other African

countries. For example, one participant stated (and was supported by majority of the participants):

*"I also know that some of them sell blood to the patients, we know it saves lives but they're selling it to the person it's supposed to save the life; so why will I go and donate if they're going to sell it to the patient."* (FGD participant 6).

Perhaps, because patients in the Ghana healthcare system are requested to pay processing fees prior to blood transfusion, the perception that donated blood are sold by hospital staff [13,31] seems not to be abating and should be holistically addressed. Furthermore, in the general questionnaire responses, nearly two-in-ten of participants suggested that blood donation was inherently associated with increased risk of contracting diseases, a misconception which has been reported elsewhere [32,33]. These findings are perhaps not surprising considering that none of the participants with previous blood donation experience indicated receiving any donor information leaflet either prior to donation or after blood donation. These donor information leaflets are meant to educate, re-enforce as well as ensure adequacy of the information issued to blood donors. Information disclosure to those who qualify or are deferred seems to be poor. For example, one FGD participant stated that (which was supported by other participants):

*"I remember they once came to our community for blood donation, I went and they performed a test and from the test, they just told me I don't qualify. And the next time at a different place too, they just told me I don't qualify to donate and that was it. I was not told what it was that did not qualify me. It was until recently when I was at the lab that was performing full blood count (FBC) that I was told that my blood level was low. So, it was then that I reflected that it may be because of this that is why I was told that I couldn't donate."* (FGD participant 8).

Another FGD participant who had previously been deferred from blood donation also added thus:

*"I think if there was something like counseling after the practitioner says you are not eligible to donate, I could have worked on it to donate the next time"* (FGD participant 7).

Taken together information disclosure is perhaps a weak link in the blood donation process and might be fueling the misconceptions. The fact that some of these misconceptions were shared between donors and non-donor groups may be indicative that some of these prior donors might not consider re-donating blood in the future. This was evidenced by the fact that only 66.7% of our participants with previous donation indicated willingness/intention to consider donating blood in the future. Previous research on blood donor retention in sub-Saharan Africa have reported low donor retention rates [34,35] which has been attributed to the low blood donations per population in sub-Saharan Africa.

Noteworthily, in agreement with previous studies [13], lack of privacy at the blood collection center was an issue that came up strongly during the focus group discussions. Both previous blood donors and non-donors alike expressed serious concerns about the infrastructural challenges that makes it impossible to communicate in confidence with the staff at the blood collection center. With such a perceived laxity in privacy, one may not be farfetched in supposing that the prospective blood donors may not have the motivation to be forthright with their responses to sensitive items on the donor history questionnaire. This lack of privacy will also be a disincentive for individuals considering blood donations attempts for first timers, and

even be a hindrance to retaining those who successfully donated blood. In the present study, only 53.7% of participants with prior blood donation considered themselves as permanent donors and were willing to donate when required.

The health benefits of blood donation are double barreled [36,37], since both the donor and potential recipient of the blood product stands to gain. In our present study, whereas only about a-third of the participants indicated that donating blood was a means of saving lives, only four-in-ten indicated that donating blood had inherent health benefits to the blood donor. This is concerning because it is well established that in the majority of cases people donate blood out of altruism [7,38]. However, if these tertiary students do not have any knowledge that their effort in donating blood will save the life of another with medical emergency, they will be less motivated to donate blood. It was therefore not surprising that 56.2% of our participants indicated that their preferred blood donation option was family replacement donations. Even among those who had previously donated blood, two-in-ten identified their status as family replacement donors which agrees with previous findings from other African countries that most people prefer replacement donation mode [13,39]. Taken together, this represents a hurdle towards achieving the 100% voluntary non-remunerated blood donation target set by the WHO in view of the fact that voluntary blood donations represent the least at-risk donation sub-type [40]. As suggested by the FGD participants, education about the blood donation processes should be prioritized:

> ". . . so, let's say in the University like those that're in the health sector that knows about blood groups can go should have some particular days that they go to the halls to educate people about it and give them knowledge about what goes into donating blood, what you should do and not do and people will get the knowledge and start coming to donate" (FGD participant 1). "We can also use influential people who are knowledgeable in blood donation to educate people on it" (FGD participant 2).

Such educational campaigns are more likely to create the needed community engagement to ensure improved donation rate per population as captured by a participant in the FGD:

> "I feel like there should be some connection with the people. Let's say it's Tafo hospital and you want to have people donating blood, you should also have something to be given to the community so let's say once in a while you go to the community and educate them on let's say disease A or B for free and let the people come out with their questions. There will be that friendship between the hospital and the community so they feel at ease to donate when you tell them to do so." (FGD participant 4)

We acknowledge that the fact that only one tertiary institution constituted our sampling frame means that our findings may not necessarily be representative of all universities in Ghana. Additionally, in view of the face-to-face focus group discussion that followed the initial questionnaire data collection, sandwich students and distance learning students were excluded from the study; this might have potentially impacted the generalizability of the research outcomes. Moreover, it is also possible that the non-probability sampling technique employed in this study might have potentially led to sampling bias since the data may reflect the experiences and views of those who are more receptive to research. However, when the generally poor knowledge about blood donation practices among these tertiary students are considered in the light of other previous studies in Ghana, our present findings are highlighting that irrespective of the educational attainment and the passage of time, knowledge about blood donation practices among Ghanaians are not improving and requires urgent attention.

## Supporting information

**S1 File. Focus group discussion 1 & 2.**
(DOCX)

**S2 File. Questionnaire data.**
(XLSX)

## Acknowledgments

We are immensely grateful to all the students who agreed to volunteer for this study. We are also grateful to the Department of Medical Laboratory Science, University of Cape Coast and the School of Allied Health Sciences, University of Cape Coast for their generosity in allowing us to use the SAHS Conference room for the focus group discussion.

## Author Contributions

**Conceptualization:** Joseph Boachie, Patrick Adu.

**Data curation:** Belinda Baidoo, Elizabeth Ankomah, Mohammed Alhassan, Godfred Benya, Emmanuella Obike, Audrey Benfo.

**Formal analysis:** Belinda Baidoo, Elizabeth Ankomah, Patrick Adu.

**Investigation:** Belinda Baidoo, Elizabeth Ankomah, Mohammed Alhassan, Godfred Benya, Emmanuella Obike, Audrey Benfo.

**Methodology:** Belinda Baidoo, Elizabeth Ankomah, Joseph Boachie.

**Project administration:** Patrick Adu.

**Resources:** Belinda Baidoo, Mohammed Alhassan, Godfred Benya, Emmanuella Obike, Audrey Benfo.

**Supervision:** Joseph Boachie, Patrick Adu.

**Validation:** Belinda Baidoo, Emmanuella Obike, Audrey Benfo, Joseph Boachie.

**Writing – original draft:** Patrick Adu.

**Writing – review & editing:** Belinda Baidoo, Elizabeth Ankomah, Mohammed Alhassan, Godfred Benya, Emmanuella Obike, Audrey Benfo, Joseph Boachie.

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
