## [Decision Letter · Decision Letter 0]

30 Aug 2023

PONE-D-23-21319Mixed-methods exploration of the knowledge of young adults about blood donation processes; a one-center cross-sectional study in a tertiary institutionPLOS ONE

Dear Dr. Adu,

Thank you for submitting your manuscript to PLOS ONE. After careful consideration, we feel that it has merit but does not fully meet PLOS ONE’s publication criteria as it currently stands. Therefore, we invite you to submit a revised version of the manuscript that addresses the points raised during the review process.

We look forward to receiving your revised manuscript.

Kind regards,

Enoch Aninagyei, PhD

Academic Editor

PLOS ONE

Journal Requirements:

Reviewers' comments:

Reviewer's Responses to Questions

**Comments to the Author**

1. Is the manuscript technically sound, and do the data support the conclusions?

Reviewer #1: Yes

Reviewer #2: Yes

Reviewer #3: Yes

2. Has the statistical analysis been performed appropriately and rigorously? 

Reviewer #1: I Don't Know

Reviewer #2: I Don't Know

Reviewer #3: I Don't Know

3. Have the authors made all data underlying the findings in their manuscript fully available?

Reviewer #1: Yes

Reviewer #2: Yes

Reviewer #3: Yes

4. Is the manuscript presented in an intelligible fashion and written in standard English?

Reviewer #1: Yes

Reviewer #2: No

Reviewer #3: Yes

5. Review Comments to the Author

Reviewer #1: Despite the fact that it is a one-center study with a limited sample size, the manuscript is technically and grammatically good. But I'm not convinced by the study's rationale. Meeting the annual blood requirement of a country directed donation is also crucial, as is replacement donation. Voluntary blood donation requires 'Education'; in other circumstances, 'Motivation' is required. Unfortunately, this study is just concerned with "Education".I have no idea about statistical analysis.

Reviewer #2: Blood donation is a very important issue, but in addition to the usual need to receive blood, malaria has added an additional burden in Africa. Considering the low amount of public aid in Ghana and the lack of response to the existing needs, the investigation of this issue is of particular importance. As mentioned in this article, the lack of sufficient information about the accepted criteria for donor selection, the effects of donation on the body's metabolism, and the lack of privacy in blood collection centers are among the things that reduce the enthusiasm of people to donate blood.

Despite revealing valuable findings, this research has not mentioned previous similar research in this field. Also, the discussion section contains a lot of unnecessary information, including quotes from participants, which could have been removed. To increase the quality of the article, I suggest rewriting the introduction and discussion using more relevant articles.

The text is clearly written but has minor grammatical problems. For example, in the second and subsequent repetitions, use the abbreviated form "SSA" and mention the word "WHO" in full in the first place.

Reviewer #3: Comments to the Author

Detailed Comments for transmission to the authors.

In this paper title ‘Mixed-methods exploration of the knowledge of young adults about blood donation processes; a one-center cross-sectional study in a tertiary institution’: Belinda Baidoo and colleagues evaluate the knowledge of university students about blood donation processes with the aim of exploring the perspectives of young adults in a tertiary educational setting to assess their knowledge regarding the blood donation requirements and procedures. The study included 382 students (18 – 49 years) studying at the University of Cape Coast (UCC) in the central region of Ghana. The students’ knowledge, perspectives and opinions about blood donation processes were assessed using a semi-structured questionnaire and focus group discussion. In their conclusions, the authors highlighted that irrespective of the educational attainment and the passage of time, knowledge about blood donation practices among Ghanaians are not improving and requires urgent attention. This is an interesting study addressing clinically relevant issues of lack of knowledge about blood donation practices among population and the necessity of creating awareness among the community to minimize the global shortage of blood donation. The following few points however deserve attention.

1. In the Background section (page ‘3’ line 31), sub-Sharan Africa (SSA). Probable a Typo mistake, if so, needs to be corrected.

2. In the material and methods/ Inclusion/exclusion criteria section (page ‘5’ line 78-81), and in the sampling of participants section (page ‘6’ line 97-98). The authors stated that regular undergraduate and post graduate UCC students were included in this study, and that others students groups (sandwich and distance learning students) were excluded. It would be better if the authors could discuss in the manuscript the reason(s) of this inclusion/exclusion criteria.

3. In the material and methods/sample size section (page ‘5’ line 91-92), the authors stated that “a total of 420 questionnaires (10% allowance) were distributed to students” whereas, in the Figure 1 (page ‘7’): The authors indicated that “424 questionnaires were distributed recruited participants”. This discrepancy needs to be rectified.

4. In the material and methods/ data collection section (page ‘7’ figure 1 [A schematic of the procedures for the data collection]), in the figure, that total number of questionnaires that were filled/not returned (382 +20 +18) does not add up to the total number of questionnaires distributed (424): this needs to be corrected.

5. In the results section (page ‘10’ table 1 [The socio-demographics of study participants]), the students age ranged from 15 - 49 years, however, the age of recruited 382 young adults stated in the abstract (page’2’ line 10-11) is 18 – 49 years. This need to be rectified.

6. PLOS authors have the option to publish the peer review history of their article (what does this mean?). If published, this will include your full peer review and any attached files.

Reviewer #1: **Yes: **Sushanta Kumar Basak

Reviewer #2: No

Reviewer #3: **Yes: **Dr Rashid Al Ghaithi

---

## [Author Response · Author response to Decision Letter 0]

10 Oct 2023

Point-by-point responses to reviewers comments as well as Editor's comments have been uploaded.

---

## [Editor Report · Decision Letter 1]

10 Nov 2023

PONE-D-23-21319R1Mixed-methods exploration of the knowledge of young adults about blood donation processes; a one-center cross-sectional study in a tertiary institutionPLOS ONE

Dear Dr. Adu,

Thank you for submitting your manuscript to PLOS ONE. After careful consideration, we feel that it has merit but does not fully meet PLOS ONE’s publication criteria as it currently stands. Therefore, we invite you to submit a revised version of the manuscript that addresses the points raised during the review process.

ACADEMIC EDITOR:Abstract: Please rewrite the abstract in an unstructured form.Lines 7,8: Revise as follows: (semi-structured questionnaire and focus group discussion [FGD])Line 53: Please change northern to NorthernLine 77: revise to Study site and populationLine 78: CentralLines 81-4: Please check, the total student breakdown doesn’t add up to 74,720Line 113: Please check, the breakdown for the 424 questionnaires distributed did not add up to the 424. Also check the first box, there is some typos. Additionally, un-highlight the 22Line 132/3: change … ‘since all the participants were tertiary students’ to … ‘since all the participants understand and speak the English language fluently’Line 374: Indicate the full form of FBC, on first mention==============================

We look forward to receiving your revised manuscript.

Kind regards,

Enoch Aninagyei, PhD

Academic Editor

PLOS ONE
---

## [Author Response · Author response to Decision Letter 1]

20 Nov 2023

All the comments raised by the Editor have been addressed as tabulated in the attached point-by-point responses to reviewers comments.

Comment Response

1. Abstract: Please rewrite the abstract in an unstructured form. Abstract has been written in an unstructured form

2. Lines 7,8: Revise as follows: (semi-structured questionnaire and focus group discussion [FGD]) Done.

3. Line 53: Please change northern to Northern Done.

4. Line 77: revise to Study site and population Done.

5. Line 78: Central Done.

6. Lines 81-4: Please check, the total student breakdown doesn’t add up to 74,720 Total number has been revised to read 74,710 in agreement with the composite numbers. The revised manuscript thus read: “The school had a total population of 74,710 of which 18,949 were regular undergraduate students, 1,445 were sandwich undergraduate students, 1,014 regular postgraduate students, 2,773 sandwich postgraduate students, 48,989 distance undergraduate students and 1,540 post graduate distance students.”

7. Line 113: Please check, the breakdown for the 424 questionnaires distributed did not add up to the 424. Also check the first box, there is some typos. Additionally, un-highlight the 22 Noted.

The numbers in the figure have been revised to correctly read “382 completely filled questionnaires, + 24 questionnaires not returned + 18 questionnaires not completely filled.” The highlight text has been removed.

8. Line 132/3: change … ‘since all the participants were tertiary students’ to … ‘since all the participants understand and speak the English language fluently’ Done.

The revised manuscript thus read “Each FGD session lasted about 60 minutes and was conducted in English language since all the participants understand and speak English language fluently” 

9. Line 374: Indicate the full form of FBC, on first mention Done.

The FBC has been defined as “full blood count”

---

## [Editor Report · Decision Letter 2]

27 Nov 2023

Mixed-methods exploration of the knowledge of young adults about blood donation processes; a one-center cross-sectional study in a tertiary institution

PONE-D-23-21319R2

Dear Dr. Adu,

We’re pleased to inform you that your manuscript has been judged scientifically suitable for publication and will be formally accepted for publication once it meets all outstanding technical requirements.

Kind regards,

Enoch Aninagyei, PhD

Academic Editor

PLOS ONE
---

## [Editor Report · Acceptance letter]

21 Dec 2023

PONE-D-23-21319R2 

PLOS ONE

Dear Dr. Adu, 

I'm pleased to inform you that your manuscript has been deemed suitable for publication in PLOS ONE. Congratulations! Your manuscript is now being handed over to our production team.

Kind regards, 

on behalf of

Dr Enoch Aninagyei 

Academic Editor

PLOS ONE